# IMF: Interpretable Multi-Hop Forecasting on Temporal Knowledge Graphs

**DOI:** 10.3390/e25040666

**Published:** 2023-04-16

**Authors:** Zhenyu Du, Lingzhi Qu, Zongwei Liang, Keju Huang, Lin Cui, Zhiyang Gao

**Affiliations:** College of Electronic Engineering, National University of Defense Technology, Hefei 230037, China; dzy17@nudt.edu.cn (Z.D.);

**Keywords:** temporal knowledge graphs, forecasting, interpretable reasoning

## Abstract

Temporal knowledge graphs (KGs) have recently attracted increasing attention. The temporal KG forecasting task, which plays a crucial role in such applications as event prediction, predicts future links based on historical facts. However, current studies pay scant attention to the following two aspects. First, the interpretability of current models is manifested in providing reasoning paths, which is an essential property of path-based models. However, the comparison of reasoning paths in these models is operated in a black-box fashion. Moreover, contemporary models utilize separate networks to evaluate paths at different hops. Although the network for each hop has the same architecture, each network achieves different parameters for better performance. Different parameters cause identical semantics to have different scores, so models cannot measure identical semantics at different hops equally. Inspired by the observation that reasoning based on multi-hop paths is akin to answering questions step by step, this paper designs an Interpretable Multi-Hop Reasoning (IMR) framework based on consistent basic models for temporal KG forecasting. IMR transforms reasoning based on path searching into stepwise question answering. In addition, IMR develops three indicators according to the characteristics of temporal KGs and reasoning paths: the question matching degree, answer completion level, and path confidence. IMR can uniformly integrate paths of different hops according to the same criteria; IMR can provide the reasoning paths similarly to other interpretable models and further explain the basis for path comparison. We instantiate the framework based on common embedding models such as TransE, RotatE, and ComplEx. While being more explainable, these instantiated models achieve state-of-the-art performance against previous models on four baseline datasets.

## 1. Introduction

Knowledge graphs (KGs) are collections of triples, such as Freebase [1] and YAGO [2]. Temporal KGs introduce a new dimension into static knowledge graphs [3], i.e., a timestamp for each triple to form a quadruple. Although there are billions of triples in temporal KGs, they are still incomplete. These incomplete knowledge bases will lead to limitations in practical applications. Since temporal KGs involve the time dimension, the completion of temporal KGs can be divided into interpolation and forecasting. The former utilizes the facts of all timestamps to predict the triples at a particular moment; the latter employs historical facts to predict future triples. Due to the importance of temporal KG forecasting in event prediction, it has attracted growing attention recently. This paper mainly focuses on temporal KG forecasting.

Most current research on temporal KG completion focuses on interpolation [4,5,6,7,8,9,10]. Recently, there have been attempts to investigate temporal KG forecasting [3,4,7,11,12,13]. According to the interpretability, research on temporal KG forecasting can be divided into two categories. One type is the black-box model, which designs an unexplainable scoring function for quadruples’ rationality. The other type is interpretable approaches. CyGNet [11] utilizes one-hop repetitive facts to realize prediction. Its performance is limited by the lack of direct repetitive knowledge of historical moments. xERTR [7], CluSTeR [3], and TITer [14] are all path-based temporal KG forecasting models. xERTR [7] adopts the inference subgraphs to aggregate local information around the question. CluSTeR [3] and TITer [14] manipulate reinforcement learning for the path search and improve the performance through temporal reasoning.

Thus far, however, there has been little discussion on the following two aspects. Firstly, uniformly measuring the paths of different hops requires handling the same semantics equivalently at different hops. Current models utilize separate networks to evaluate paths at different hops. Although each hop’s network has the same architecture, each network acquires different parameters for better performance. Different parameters cause identical semantics to have different scores, so current models cannot truly compare multi-hop paths according to the same criteria. For example, xERTR [7] simply gathers the scores of different paths for comparison, which is mainly based on training datasets. Secondly, although current models can provide reasoning paths, the comparison of paths operates in a black-box fashion. The interpretability of the current models means providing the reasoning paths, which is an essential property of path-based models. These models lack an explanation of the preference for various paths, i.e., they cannot provide the basis for path comparison.

In practice, forecasting based on path searching aims to find the appropriate multi-hop paths, the combination of whose relations is equivalent to the question’s relation. As we observe, reasoning based on multi-hop paths is akin to stepwise question answering. Inspired by stepwise question answering, this paper designs a new Interpretable Multi-Hop Reasoning (IMR) framework based on consistent basic models, which can uniformly integrate the paths of different hops and perform more interpretable reasoning.

The primary pathway of IMR can be as follows. IMR first transforms reasoning based on path searching into stepwise question answering based on basic KG embedding models [1,15,16,17,18] and IRN [19]. This framework calculates the unanswered parts of questions after each hop as the new question for the next hop during the stepwise question answering, which is named the remainder of questions in this paper. Moreover, IMR designs three indicators based on the unanswered parts of questions and the inferred tails: the query matching degree, answer completion level, and path confidence. The query matching degree, i.e., the matching degree between the reasoning tails and the original questions, measures the rationality of the new quadruples. The answer completion level, i.e., the matching degree between the relations of paths and that of the questions, measures the answer’s completeness. Path confidence, i.e., the difference between the same entities with different timestamps, measures the reliability of the reasoning paths. IMR achieves the unified scoring of multi-hop paths and better explainable reasoning simultaneously with these indicators’ combination.

The major contributions of this work are as follows. (1) A new Interpretable Multi-Hop Reasoning framework (IMR) is proposed in this paper, which provides a new framework for the specific design of forecasting models. Furthermore, IMR defines three indicators: the query matching degree, answer completion level, and path confidence. (2) Unlike other models that cannot measure the paths of different hops uniformly, IMR can measure the paths of different hops according to the same criteria and utilize multi-hop paths for inference. (3) IMR can provide reasoning paths similarly to other interpretable models and further explain the basis for path comparison. (4) Based on basic embedding models, IMR is instantiated as the specific model. Experiments on four benchmark datasets show that these instantiated models achieve state-of-the-art performance against previous models.

## 2. Related Work

**Static KG reasoning.** Knowledge graph reasoning based on representation learning has been widely investigated by scholars. These approaches to reasoning can be categorized into geometric models [1,17,20,21,22], tensor decomposition models [15,16,18,23], and deep learning models [24,25,26]. In recent years, some scholars have attempted to introduce GCN into knowledge graph reasoning [27], which can improve the performance of basic models. Some other scholars focus on multi-hop reasoning with symbolic inference rules learned from relation paths [28,29]. The above methods are all designed for static KGs, making it challenging to deal with temporal KG reasoning.

**Temporal KG reasoning.** Temporal KGs import the time dimension into static KGs, which makes the facts of a specific timestamp extremely sparse. The temporal KG reasoning task can be divided into two categories: reasoning about historical facts [4,5,6,7,8,30], i.e., interpolation on temporal KGs, and reasoning about future facts [3,4,7,11], i.e., forecasting on temporal KGs. The former predicts the missing facts of a specific historical moment based on the facts of all moments, and the latter predicts future events based only on the past facts. There are many studies on the task of temporal KG interpolation. However, these studies are all black-box models, which cannot explain predictions. Most of the proposed models for temporal KG forecasting are also black-box models. BoxTE [31] utilizes BoxEmbedding for temporal KG forecasting, which is expressive and possesses an inductive capacity. Recently, xERTR [7], CluSTer [3], and TITer [14] were shown to explain predictions to some extent. These models can provide the reasoning paths for the predictions. However, both models cannot truly handle multi-hop paths crossing the same criteria, which is more similar to the weighted combination. xERTR and TiTer combine the scores of paths with different hops by training weights. Experiments show that CluSTeR performs worse on paths with multiple hops than on paths with only one hop.

Most current temporal KG forecasting models are black-box models. Only some models can provide reasoning paths for prediction. Moreover, none of them can explain how path comparisons work and none of them can integrate paths of different hops uniformly.

## 3. Preliminaries

**The task of temporal KG forecasting.** Suppose that E, R, and T represent the entity set, predicate set, and timestamp set, respectively. The temporal KG is a collection of quadruples, which can be expressed as
(1)$$K=\left\{\left(e_{s},r,e_{o},t\right),e_{s},e_{o}\in E,r\in R,t\in T\right\}$$

$\left(e_{s},r,e_{o},t\right)$ denotes a quadruple; $e_{s}$ and $e_{o}$ represent the subject and object, respectively. *r* represents the relation, and *t* represents the time that the quadruple occurs. Suppose that facts happening before the selected time $t_{k}$ can be expressed as
(2)$$G_{t_{k}}=\left\{\left(e_{i},r,e_{j},t_{i}\right)\in K|t_{i}<t_{k}\right\}$$

Temporal KG forecasting predicts future links based on past facts. This means that its foundation is the process of predicting $e_{o}$ based on a question $\left(e_{s},r_{q},?,t_{q}\right)$ and the previous facts $G_{t_{q}}$, where $r_{q},t_{q}$ denote the relation and timestamp of the question. Temporal KG forecasting involves ranking all entities of the specific moment and obtaining the preference for prediction.

**Temporal KG forecasting based on paths.** Knowledge graph embedding associates the entities e∈E and relations r∈R with vectors e,r. Different from static KGs, the entities in temporal KGs contain time information. The entity may contain different attributes at different moments. In order to better characterize the entity in temporal KGs, we associate each entity *e* with a specific time label $t_{i}\in T$, so the entity *e* can be depicted as $e^{t_{i}}$ and its embedding can be denoted as $e^{t_{i}}$. The set of quadruples directly associated with $e_{s}^{t_{i}}$, which can be defined as the 1-hop paths associated with $e_{s}^{t_{i}}$, can be expressed as $P_{\left(e_{s},t_{i}\right)}=\left\{\left(e_{s},r,e_{j},t_{k}\right)|\left(e_{s},r,e_{j},t_{k}\right)\in G_{t_{i}}\right\}$, where $e_{s},e_{j}\in E,r_{p}\in R,t_{k}<t_{i}\in T$. In this way, $P_{\left(e_{s},t_{i}\right)}$ can represent all associated quadruples. The set of entities directly associated with $e_{s}^{t_{q}}$ in the path $P_{\left(e_{s},t_{q}\right)}$, i.e., the 1-hop neighbors of $e_{s}^{t_{q}}$, can be denoted as $N_{\left(e_{s},t_{q}\right)}=\left\{e_{i}^{t_{h}}|\left(e_{s},r,e_{i},t_{h}\right)\in P_{\left(e_{s},t_{q}\right)}\right\}$, where $e_{s},e_{i}\in E,r\in R,t_{h}<t_{q}\in T$. Given the question $\left(e_{s},r_{q},?,t_{q}\right)$, the forecasting task can be depicted as requesting the entity $e_{o}$ based on path searching. For example, we search the path with $e_{s}$ as the starting point:(3)$$\left(e_{s},r_{p_{\left(1\right)}},e_{1},t_{1}\right),\left(e_{1},r_{p_{\left(2\right)}},e_{2},t_{2}\right),\ldots ,\left(e_{i-1},r_{p_{\left(i\right)}},e_{i},t_{i}\right)$$
where $r_{p_{\left(i\right)}}$ denotes the relations of the *i*th-hop. Thus, answers to the question may be $e_{1},e_{2},e_{3},\ldots ,e_{i},$ and the corresponding inference hop is 1,2,3,…,i, respectively. Moreover, $e_{s_{\left(i\right)}},r_{q_{\left(i\right)}}$ denotes the remaining (or unanswered) subjects and relations of questions after the *i*th-hop paths, which will be explained in Section 4.3.2.

**Uniformly measuring paths of different hops.** Uniformly measuring paths of different hops requires models scoring paths of different hops according to the same criteria. For example, given question $\left(e_{s},r_{q},?,t_{q}\right)$ and the searched 1-hop path $\left(e_{s},r_{p},e_{1},t_{1}\right)$, the score obtained for the searched 1-hop path is *f*. If we find no path during the first hop, the original question is left to the second hop to solve. Thus, the remaining question (unanswered question) for the second hop is still $\left(e_{s},r_{q},?,t_{q}\right)$. When the path searched at the second hop is also $\left(e_{s},r_{p},e_{1},t_{1}\right)$, the score for the searched path at the second hop should also be *f*. As is shown in this example, we should score identical semantics equivalently even under different hops. Moreover, the equal comparison of paths provides the basis for the interpretability of path comparison. This attribute constrains models to have an identical scoring mechanism at each hop, i.e., each hop’s separate networks for the models based on neural networks should have the same parameters. However, only IMR can meet the attribute.

**Fact matching based on TransE.** This paper is the first study of the design of interpretable evaluation indicators from the perspective of actual semantics. We instantiate IMR to better illustrate the design pathway and thus choose the basic embedding model TransE as the basis of IMR. In TransE, relations are represented as translations in the embedding space. If the triple $\left(e_{s},r,e_{o}\right)$ holds in static KGs, TransE [1] assumes the following relationship.
(4)$$\left|e_{s}+r-e_{o}\right|=0$$
where $e_{s},r$ and $e_{o}\in R^{k}$, and *k* denotes the dimension of each vector.

For each quadruple $\left(e_{s},r_{q},e_{o},t_{q}\right)$ in temporal KGs, the relation $r_{q}$ can also be taken as the translation from the subject $e_{s}$ to the object $e_{o}$, i.e., $e_{s}^{t_{q}}+r_{q}=e_{o}^{t_{q}}$. We suppose that when the distance *d* of quadruples is smaller, the quadruple will be better matched. The distance of the quadruple $\left(e_{s},r_{q},e_{o},t_{q}\right)$ can be expressed as
(5)$$d=\left|e_{s}^{t_{q}}+r_{q}-e_{o}^{t_{q}}\right|$$

The relations in KG embedding models indicate the translations between entities, whose specific design determines the complexity of the indicators designed by IMR. The design route of IMR originates from the perspective of reasoning from actual semantics, which is not limited to specific basic models. The consistent basic model of IMR-TransE is TransE, i.e., all IMR-TransE’s specific formulas are based on TransE, which will not be explained below. To limit the length of the paper, we move the details of IMR-TransE and IMR-ComplEx to Section A.2.

## 4. IMR: Interpretable Multi-Hop Reasoning

We introduce the Interpretable Multi-Hop Reasoning framework (IMR) in this section. We first provide an overview of IMR in Section 4.1. IMR comprises three modules: the path searching module, query updating module, and path scoring module. The path searching module searches related paths hop by hop from the subjects of questions, involving path sampling and entity clipping, whose motivation and design are presented in Section 4.2. The query updating module calculates the remaining questions hop-by-hop for each path, involving the update of the subject and relations, whose motivation and design are introduced in Section 4.3. The path scoring module designs three indicators: the question matching degree, answer completion level, and path confidence. This module combines three indicators to evaluate each path, whose motivation and design are presented in Section 4.4. We introduce training strategies and the regularizations on state continuity in Section 4.5. IMR conducts uniform path comparisons based on consistent basic models. To better illustrate this framework, we also include the corresponding instance model (IMR-TransE) in Section 4.3, Section 4.4 and Section 4.5. The detailed implementations of IMR-RotatE and IMR-ComplEx are included in Section A.2.

### 4.1. Framework Overview

We notice that predicting unknown facts based on paths is akin to answering questions, i.e., the question can be answered directly via finding triples with an equal relation or gradually by utilizing the multi-hop equivalent paths. Inspired by this observation, we take the task of link prediction as stepwise question answering. IMR primarily consists of searching for paths hop by hop, updating the remaining questions for each path, and filtering the best answers based on three indicators: the question matching degree, answer completion level, and path confidence.

We show a toy example in Figure 1. Given a question $\left(e_{s},r_{q},?,t_{q}\right)$ and the previous facts $G_{t_{q}}$, the task of forecasting is predicting the missing object $e_{o}$. The steps of IMR are as follows.

**Step 1**: Starting from the subject $e_{s}$, we first acquire the associated quadruples $P_{\left(e_{s},t_{q}\right)}$, namely 1-hop paths. We temporally bias the neighborhood sampling using an exponential distribution for the neighbors [7]. The distribution negatively correlates with the time difference between node $e_{s}$ and its neighbor $N_{\left(e_{s},t_{q}\right)}$. Then, we calculate the remaining questions (the remaining subject $e_{s_{\left(1\right)}}$ and the remaining relation $r_{q_{\left(1\right)}}$) for each sampled path. Finally, IMR scores 1-hop paths based on three indicators, which is discussed in Section 4.4.

**Step 2**: To prevent the path searching from exploding, the model samples the tails of 1-hop paths for the 2-hop path searching. As shown by the pink arrow in Figure 1, the tails of 1-hop paths are clipped according to the scores of 1-hop paths. For the 2-hop paths searched from the clipped tails, IMR samples the paths negatively correlated with time distances. Then, IMR calculates the remaining questions for each 2-hop path (the remaining subject $e_{s_{\left(2\right)}}$ and the remaining relation $r_{q_{\left(2\right)}}$) and scores the 2-hop paths based on three indicators.

**Step 3**: Rank the scores of 1-hop and 2-hop paths to obtain the preference answer.

### 4.2. Path Searching Module

Inspired by the observation that reasoning based on multi-hop paths is akin to stepwise question answering, this module searches related paths hop by hop from the subjects of questions.

**Path sampling.** For the path searching from the starting subject $e_{s}^{t_{q}}$, the number of triples in $P_{\left(e_{s},t_{q}\right)}$ may be very large. To prevent the path searching from exploding, we sample a subset of the paths. In fact, the attributes of entities in temporal KGs may change over time. Consider the observation that when $t_{1}$ is closer to $t_{q}$, the attributes of $e_{s}^{t_{1}}$ should be more similar to those of $e_{s}^{t_{q}}$. We also verify the correlation between attributes and the time distance in Section A.6. Therefore, we are more prone to sample nodes whose time is closer to $t_{q}$. In this paper, we employ time-aware exponentially weighted sampling in xERTR [7]. xERTR temporally biases the neighborhood sampling using an exponential distribution of temporal distance.

**Entity pruning.** The search for next-hop paths is based on the tails of previous-hop paths, so the number of paths is increased by the exponent of dimensions. To avoid the explosion of next-hop path searching, this paper proposes to select the top-K entities for the next-hop search based on the sorted scores of the previous hops.

### 4.3. Query Updating Module

Given a question $\left(e_{s},r_{q},?,t_{q}\right)$, there may be a few relations directly equivalent to $r_{q}$ in the temporal KGs for the task of link prediction. More questions need to go through multi-hop paths to infer the outcome. In question answering, a complex question can be decomposed into multiple sub-questions, with one sub-question answered at each step. Thus, inference based on the multi-hop path is equivalent to answering complex questions step by step. Moreover, we need to remove the part resolved to focus on the remaining questions. IMR proposes to update the question according to the last hop and focus on finding the unsolved parts. The query updating module mainly calculates the remaining questions, i.e., the unanswered questions.

The embedding of entities is first introduced in this subsection, followed by the query updating module of IMR-TransE.

#### 4.3.1. Entity Representation

The attributes contained in the entities may change over time. This paper divides the entity embeddings of each timestamp into a static representation and dynamic representation.
(6)$$e=act\left(MLP([\;e_{\mathrm{sta}}\;||\;e_{\mathrm{dy}}\;])\right)$$

Here, the vector $e_{\mathrm{sta}}$ denotes the static embedding, which captures time-invariant features and global dependencies over the temporal KGs. The vector $e_{\mathrm{dy}}$ represents the dynamic embedding for each entity that changes over time. || denotes the operation of concatenation and MLP(·) denotes the multilayer perceptron (MLP). act(·) denotes the activation function. We provide more details about $e_{\mathrm{sta}}$ and $e_{\mathrm{dy}}$ in Section A.3.

#### 4.3.2. Question Updating

Each path contains a different set of relations. After each hop, the question needs to discard the processed semantic, i.e., to obtain the remaining subject and relation of the question.

**Question updating for IMR-TransE.** As shown in Figure 1, the subject and relation of the question after the *i*-th hop path are updated based on Equation (Equation 5) as follows.
(7)$$e_{s_{\left(i\right)}}=e_{s_{\left(i-1\right)}}+r_{p_{\left(i\right)}}$$
(8)$$r_{q_{\left(i\right)}}=r_{q_{\left(i-1\right)}}-r_{p_{\left(i\right)}}$$
where the embedding $e_{s_{\left(i\right)}}$ and $r_{q_{\left(i\right)}}$ represent the remaining subject and relation of the question after the *i*-hop path, respectively. Moreover, $e_{s_{\left(0\right)}}=e_{s}$, $r_{q_{\left(0\right)}}=e_{q}$ and $r_{p_{\left(i\right)}}$ denotes the relation of *i*-th hop path and *i* is the number of hops for each path.

### 4.4. Path Scoring Module

For the question (Sub, Rel, ?, Tq), we search the 2-hop path (Sub, R1, Obj1, T1),(Obj1, R2, Obj2, T2). The pink box indicates that the original question and the tail of the path are combined as a quadruple to measure the rationality of searched tails, i.e., the question matching degree $f_{qmd}$. The purple box represents the comparison between the question’s relation and the path relations to measure the semantic equivalence between the question and the path, i.e., the answer completion level $f_{ac}$. These green boxes compare the attributes of the same entities with different timestamps to measure the reliability of the search path, i.e., the path confidence $f_{pc}$.

We evaluate the path searching from three perspectives. First, the searched tails should match the original question, which means that the correct tails searched by paths and the question should satisfy the consistent basic embedding model. Secondly, the ideal path should be the search for equivalent semantics for relations, not merely the search for the correct tails. It is necessary to ensure the correctness of semantic equivalence, i.e., the path is semantically equivalent to the relation of the question. Finally, considering the particularity of the temporal KGs, the attributes of the same entity may change over time. The current sampling strategy for path searching is to sample adjacent timestamp triples of the same entity. When the attribute value of the entity changes significantly over time, it is inappropriate to perform this sampling strategy for the next hop. We need to ensure that the same entity with different timestamps has similar properties in the same path. In this way, three indicators have been developed by IMR to measure the rationality of the reasoning path, respectively: the question matching degree, answer completion level, and path confidence. Although the current methods, such as models based on reinforcement learning, can have complicated designs, the score functions simply belong to a type of question matching degree. We provide a detailed analysis of the correlation between IMR and reinforcement-learning-based models in Section A.5.

#### 4.4.1. Question Matching Degree

For the tails found by path searching, we need to measure the matching degree between the tails and the question, the question matching degree. In fact, the scoring function applied by some traditional reinforcement learning methods is a type of question matching degree. As shown in the yellow box in Figure 2, for the entity $e_{p_{\left(i\right)}}^{t_{i}}$ searched by the paths with *i* hops, we combine the entity $e_{i}^{t_{i}}$ and the question $\left(e_{s},r_{q},?,t_{q}\right)$ into a new quadruple $\left(e_{s},r_{q},e_{p_{\left(i\right)}}^{t_{i}},t_{q}\right)$.

**Question matching degree for IMR-TransE.** Question matching degree $f_{qmd}$ in IMR-TransE calculates the distance of the constructed quadruple based on TransE [1]. The better the entity matches the question, the smaller the distance of quadruples will be. The calculation of $f_{qmd}$ for *i*th-hop path is as follows.
(9)$$f_{qmd}^{i}={∥e_{s}^{t_{q}}+r_{q}-e_{p_{\left(i\right)}}^{t_{i}}∥}_{p}$$
where the *p*-norm of a complex vector *V* is defined as ${∥V∥}_{p}=\sqrt[{p}]{{\left|V_{i}\right|}^{p}}$. We use the L1-norm for all indicators in the following.

#### 4.4.2. Answer Completion Level

Among the paths to the right tails, some paths are not related to the semantics of the question. Although these paths can infer the tail, these paths are invalid due to being unrelated to the question in semantics. Therefore, IMR designs an index to measure the semantic relevance between the path and the question. Answer completion level $f_{ac}$ indicates whether the combination of path relations can reflect the relation of the question in semantics. IMR takes the remaining relations of the question as the answer completion level, which is calculated based on the distance between the relations of paths $r_{p_{\left(1\right)}},r_{p_{\left(2\right)}},\ldots$ and the relation $r_{q}$. The fewer the relations of a question that remain, the more complete the answer given by the combination of path relations.

**Answer completion level for IMR-TransE.** The calculation of $f_{ac}$ for *i*th-hop path in IMR-TransE is as follows.
(10)$$\begin{array}{cc} f_{ac}^{i} & ={∥r_{q}-r_{p_{\left(1\right)}}-r_{p_{\left(2\right)}}-r_{p_{\left(3\right)}}-\ldots -r_{p_{\left(i\right)}}∥}_{p}\\ \; & ={∥r_{q_{\left(1\right)}}-r_{p_{\left(2\right)}}-r_{p_{\left(3\right)}}-\ldots -r_{p_{\left(i\right)}}∥}_{p}\\ \; & ={∥r_{q_{\left(i\right)}}∥}_{p} \end{array}$$

#### 4.4.3. Path Confidence

Path searching is the process of searching for the next-hop paths based on the tail of the previous hop. When searching for a path, the current sampling strategy is to sample adjacent timestamp triples of the same entity. There are deviations between the same entities with different timestamps in temporal KGs. The premise of this sampling strategy is that only when entities have similar attributes under different timestamps, the path searching is valid. When the entity’s attributes change significantly over time, performing an effective next path search is inappropriate. The reasoning path is more reliable when the deviations between entities are smaller. IMR designs path confidence $f_{pc}$, i.e., the error between the subject of the updated question $e_{s_{\left(i\right)}}$ and the tails $e_{p_{\left(i\right)}}^{t_{i}}$ of the path with *i* hops.

**Path confidence for IMR-TransE.** The calculation of $f_{pc}$ for *i*th-hop path in IMR-TransE is as follows.
(11)$$f_{pc}^{i}={∥e_{s_{\left(i\right)}}-e_{p_{\left(i\right)}}^{t_{i}}∥}_{p}$$
where $e_{q_{\left(i\right)}}$ represents the remaining subject of the question updated by paths of the length *i*, and $e_{p_{\left(i\right)}}^{t_{i}}$ represents the tail reasoned by the *i*-hop paths.

#### 4.4.4. Combination of Scores

IMR merges indicators with positive weights to obtain the final score of each path, i.e., $f=w_{qmd}*f_{qmd}+w_{ac}*f_{ac}+w_{pc}*f_{pc}$, where $w_{qmd},w_{ac},w_{pc}\in R^{+}$.

**Entity aggregation for IMR.** Considering that the searched paths may lead to entities with different timestamps, IMR adopts specific aggregation for searched entities. First, entities with the same timestamp may be inferred by different paths, so IMR needs to combine the scores of entities with unique timestamps. Considering that only one path matches the question best, IMR employs max aggregation to various paths reaching the same entities with the same timestamp. Moreover, specific paths may infer the same entity with different timestamps. IMR performs average aggregation on the scores of entities with different timestamps. Finally, IMR obtains the score of each entity at the question timestamp.

### 4.5. Learning

We utilize binary cross-entropy as the loss function, which is defined as
(12)$$\begin{array}{cc} L & =-\frac{1}{\left|Q\right|}\sum_{q\in Q}\frac{1}{\left|\varepsilon_{q}^{p}\right|}\sum_{e_{i}\in \varepsilon_{q}^{p}}\left(y_{e_{i},q}\log\left(\frac{f_{e_{i},q}}{\sum_{e_{i}\in \varepsilon_{q}^{p}}f_{e_{i},q}}\right)\right)\\ \; & +\frac{1}{\left|Q\right|}\sum_{q\in Q}\frac{1}{\left|\varepsilon_{q}^{p}\right|}\sum_{e_{i}\in \varepsilon_{q}^{p}}\left(\left(1-y_{e_{i},q}\right)\log\left(1-\frac{f_{e_{i},q}}{\sum_{e_{i}\in \varepsilon_{q}^{p}}f_{e_{i},q}}\right)\right) \end{array}$$
where $\varepsilon_{q}^{p}$ represents the set of entities reasoned by selected paths. $y_{e_{i},q}$ represents the binary label that indicates whether it is the answer for *q* and *Q* represents the training set. $f_{e_{i},q}$ denotes the score obtained by Section 4.4.4 for each path. We jointly learn the embeddings and other model parameters by end-to-end training.

**Regularization.** For the same entity with different timestamps, the closer its time distance is, the closer its dynamic embedding is [32]. IMR proposes the regularization on continuity for the dynamic vectors of entities.

The specific regularization for IMR is as follows.
(13)$$\begin{array}{c} reg={∥e_{k}^{t_{j}}-e_{k}^{t_{j-1}}∥}_{p}+{∥e_{k}^{t_{j}}-e_{k}^{t_{j+1}}∥}_{p} \end{array}$$
where $e_{k}^{t_{j}}$ denotes the dynamic embedding of the *k*-th entity at the *j*-th timestamp. $e_{k}^{t_{j-1}},e_{k}^{t_{j+1}}$ denotes the dynamic embedding of the previous and later timestamp against $e_{k}^{t_{j}}$, respectively. ${∥\cdot ∥}_{p}$ denotes the *p* norm of the vectors and we take *p* as 1 in this paper.

## 5. Experiments

### 5.1. Datasets and Baselines

To evaluate the proposed module, we consider two standard temporal KG datasets Integrated Crisis Early Warning System (ICEWS) [33], WIKI [34], and YAGO [35]. The ICEWS dataset contains information about political events with time annotations. We select two subsets of the ICEWS dataset, i.e., ICEWS14 and ICEWS18, containing event facts in 2014 and 2018. WIKI and YAGO is a temporal KG that fuses information from Wikipedia with WordNet [36]. Following the experimental settings of HyTE [37], we deal with year-level granularity by dropping the month and date information. We compare IMR and baseline methods by performing the temporal KG forecasting task on ICEWS14, ICEWS18, WIKI, and YAGO. Details of these datasets are listed in Table 1. We adopt the same dataset split strategy as in [38].

We compare the performance of IMR-TransE against the temporal KG reasoning models, including TTransE [34], TA-DistMult/TA-TransE [30], DE-SimplE [39], TNTComplEx [32], CyGNet [11], RE-Net [38], TANGO [40], TITer [14], and xERTR [7].

In the experiments, the widely used Mean Reciprocal Rank (MRR) and Hits@1,3,10 are employed as the metrics. The filtered setting for static KGs is not suitable for the reasoning task under the exploration setting, as mentioned in xERTR [7]. This paper adopts the time-aware filtering scheme, which only filters out genuine triples at the question time.

### 5.2. Experimental Results

**Main results.** Table 2 and Table 3 show the comparison between IMR-TransE, IMR-RotatE, IMR-ComplEx, and other baseline models on ICEWS, WIKI, and YAGO. Overall, the instantiated models of IMR outperform the baseline models in all metrics while being more interpretable, which convincingly verifies its effectiveness. Due to the limited paper length, a detailed analysis of the interpretability is provided in Section A.1. Compared to the best baseline (TiTer), IMR-TransE obtains a relative improvement of 3.3% and 2.5% in MRR and Hits@1, averaged on ICEWS, WIKI, and YAGO. Moreover, different IMR models achieve the best performance across unique datasets due to basic models.

**Comparison of multi-hop paths.** Figure 3 shows the performance of IMR-TransE on ICEWS, WIKI, and YAGO as the maximum length of paths increases. The performance basically continues rising with the increase in the paths’ length. However, as the maximum length of paths increases, the performance on ICEWS18 hardly improves. Further analysis on ICEWS18 in [3] explains that there are no strong dependencies between the relations of the question and the multi-hop paths. Thus, longer paths provide little gain for inference. Moreover, as the maximum length of paths increases, the number of inference paths increases exponentially and most of the invalid paths will suppress the performance of IMR-TransE. In order to ensure that the performance of the model does not decrease, we propose to control the sampling number of next-hop paths to limit the total number of multi-step paths and suppress the impact of noisy samples. This paper set the number of next-hop samplings to 5. In summary, experiments show that unified indicators designed by IMR based on consistent basic models can uniformly measure the paths of different hops, allowing better reasoning based on paths with different hops, which verifies the claim in Section 4.4. We present an extra ablation study on three indicators in IMR-TransE in Section A.4.

## 6. Conclusions

We propose an Interpretable Multi-Hop Reasoning framework for temporal KG forecasting tasks. IMR transforms reasoning based on path searching into stepwise question answering based on consistent basic models. Moreover, IMR develops three indicators to measure the answer and reasoning paths, and this is the first study to develop interpretable evaluation indicators from the perspective of actual semantics for the temporal KG forecasting task. IMR can measure the paths of different hops according to the same criteria and be more explainable. Extensive experiments on four benchmark datasets demonstrate the effectiveness of our method. In the future, we plan to enhance the prediction by integrating different paths reaching the same tail, which will be more effective and interpretable. We will also continue to explore the models based on GAT [3] for temporal KG forecasting tasks.

## Figures and Tables

**Figure 1 entropy-25-00666-f001:**
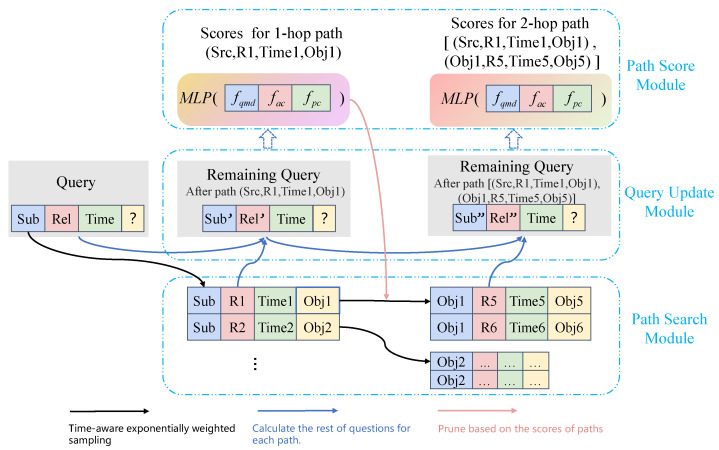
The architecture of IMR. We take the 2-hop path search as an example. The black and red arrows denote time-aware exponentially weighted sampling and pruning based on the scores of paths, respectively (Section 4.2). The blue arrows denote the calculation of the rest of the questions for each path (Section 4.3). (Sub, Rel, ?, Time) is regarded as the original question, which can be denoted as $\left(e_{s},r_{q},?,t_{q}\right)$. The searched two paths are [(Sub,R1,Obj1,Time1)] and [(Sub,R1,Obj1,Time1),(Obj1,R5,Obj5,Time5)], which can be denoted as $\left[\left(e_{s},r_{p_{\left(1\right)}},e_{1},t_{1}\right)\right]$ and $\left[\left(e_{s},r_{p_{\left(1\right)}},e_{1},t_{1}\right),\left(e_{1},r_{p_{\left(2\right)}},e_{2},t_{2}\right)\right]$, respectively. (Sub’, Rel’, ?, Time) and (Sub”, Rel”, ?, Time) denote the remaining questions after the 1-hop and 2-hop path, which can be taken as $\left(e_{s_{\left(1\right)}},r_{q_{\left(1\right)}},?,t_{q}\right),\left(e_{s_{\left(2\right)}},r_{q_{\left(2\right)}},?,t_{q}\right)$, respectively.

**Figure 2 entropy-25-00666-f002:**
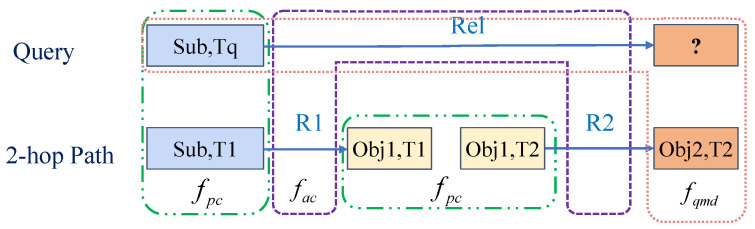
A brief illustration of the path scoring module.

**Figure 3 entropy-25-00666-f003:**
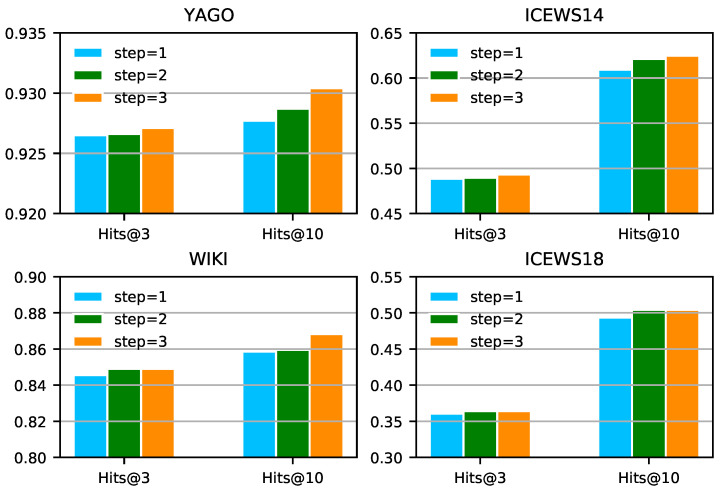
Comparison of the performance of paths with different maximum hops on four datasets. We average the output of four experiments with different random seeds and fixed hyperparameters.

**Table 1 entropy-25-00666-t001:** Statistics of three benchmark datasets.

Dataset	ICEWS14	ICEWS18	WIKI	YAGO
entity	7128	23,033	12,554	10,623
relation	230	256	24	10
timestamp	365	304	232	189
training	63,685	373,018	539,286	161,540
validation	13,823	45,995	67,538	19,523
test	13,222	49,545	63,110	20,026

**Table 2 entropy-25-00666-t002:** Results comparison on ICEWS14 and ICEWS18. Compared metrics are time-aware filtered MRR (%) and Hits@1/3/10 (%), which are multiplied by 100. The best results among all models are in bold.

	ICEWS14				ICEWS18			
	MRR	Hit@1	Hit@3	Hit@10	MRR	Hit@1	Hit@3	Hit@10
TTransE	13.43	3.11	17.32	34.55	8.31	1.92	8.56	21.89
TA-DistMult	26.47	17.09	30.22	45.41	16.75	8.61	18.41	33.59
DE-SimplE	32.67	24.43	35.69	49.11	19.30	11.53	21.86	34.80
TNTComplEx	32.12	23.35	36.03	49.13	27.54	19.52	30.80	42.869
CyGNet	32.73	23.69	36.31	50.67	24.93	15.90	28.28	42.61
RE-NET	38.28	28.68	41.34	54.52	28.81	19.05	32.44	47.51
xERTE	40.79	32.70	45.67	57.30	29.31	21.03	33.51	46.488
TANGO-Tucker	–	–	–	–	28.68	19.35	32.17	47.04
TANGO-DistMult	–	–	–	–	26.75	17.92	30.08	44.09
TITer	41.73	32.74	46.46	58.44	29.98	22.05	33.46	
IMR-TransE	**44.76**	**35.64**	**49.49**	**62.30**	32.45	22.97	36.05	49.36
IMR-RotatE	44.21	35.13	48.72	62.04	32.67	23.53	36.76	50.67
IMR-ComplEx	44.03	34.55	49.21	62.11	**33.33**	**24.07**	**37.65**	**51.51**

**Table 3 entropy-25-00666-t003:** Results comparison on WIKI and YAGO. Compared metrics are time-aware filtered MRR (%) and Hits@1/3/10 (%), which are multiplied by 100. The best results among all models are in bold.

	WIKI				YAGO			
	MRR	Hit@1	Hit@3	Hit@10	MRR	Hit@1	Hit@3	Hit@10
TTransE	29.27	21.67	34.43	42.39	31.19	18.12	40.91	51.21
TA-DistMult	44.53	39.92	48.73	51.71	54.92	48.15	59.61	66.71
DE-SimplE	45.43	42.6	47.71	49.55	54.91	51.64	57.30	60.17
TNTComplEx	45.03	40.04	49.31	52.03	57.98	52.92	61.33	66.69
CyGNet	33.89	29.06	36.10	41.86	52.07	45.36	56.12	63.77
RE-NET	49.66	46.88	51.19	53.48	58.02	53.06	61.08	66.29
xERTE	71.14	68.05	76.11	79.01	84.19	80.09	88.02	89.78
TANGO-Tucker	50.43	48.52	51.47	53.58	57.83	53.05	60.78	65.85
TANGO-DistMult	51.15	49.66	52.16	53.35	62.70	59.18	60.31	67.90
TITer	75.50	72.96	77.49	79.02	87.47	84.89	89.96	90.27
IMR-TransE	80.41	76.04	84.91	85.95	90.24	87.91	92.65	92.77
IMR-RotatE	79.43	74.36	84.59	85.79	**90.34**	**88.10**	92.69	**92.78**
IMR-ComplEx	**80.54**	**76.12**	**84.98**	**85.97**	90.19	87.80	**92.71**	92.78

## Data Availability

Not applicable.

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
