# Peer review of "IMF: Interpretable Multi-Hop Forecasting on Temporal Knowledge Graphs"

_entropy, 2023, doi:10.3390/e25040666_

Round 1
Reviewer 1 Report
Actually, the manuscript proposes a novel contribution, and it is well organized and well written. Thus, I recommend it for publication.
Author Response
Point 1:
Actually, the manuscript proposes a novel contribution, and it is well organized and well written. Thus, I recommend it for publication..
Response 1:
Thank you very much for your careful review! And if you have any questions or comments, please contact us!

Reviewer 2 Report
1. The English language needs more work. There are many grammatical and typo mistakes in this manuscript. The paper needs to be edited by a native English speaker.
2. I suggest the authors revise the introduction of the study per the comments raised. The authors can also use the following points below as a guideline to help them come out with an interesting introduction that is more scientific.
Background & Significance: (What general background does the reader need in order to understand the manuscript and how important is it in the context of scientific research).
Problem definition: (What are the research questions to fill in the gaps of the existing knowledge body or methodology (Methods are not a contribution, but a tool to assess whether your hypothesis or predictions are supported or not supported)? I would like to see well developed arguments for predicting or proposing specific relationships in this study.
Motivations & Objectives: (Why are you conducting the study and what are the goals to achieve?)
3. Explain the variable selection procedure; why the authors choose these variables for the study.
4. I would like to suggest that authors should update the introduction, literature, and results part. Specifically, the latest research trends, and in order to highlight the academic frontier of the research, the references of the recent year need to be referenced.
5. The author(s) need to compare their results (Each Findings) with past studies (what was provided in the article is not compares of results but an explanation of views from past authors) and in comparing the result from the empirical investigations the author(s) should as much as possible provide a recast of the comparison made and the supposed implications or advantages of the new finding made with those discovered by past authors. This will ensure justice to the extant literature and also evincing the superiority of the current findings over the past findings.
6. The conclusion of the study should have a realistic empirical overview and not a summary. The conclusion should provide an overall thought from the author(s) empirical and conceptual viewpoint on why and how things exist or went the way they were discovered, what are the implications of that and what are the advantages to the key study areas under survey. Is this development attributable to the method or the variables or are these findings the reality of the situations on the ground or what is to be expected soonest etc? This should be presented in a professional, logical, and philosophical way to convey some key scientific thoughts to the readers. The present conclusion provided in the manuscript needs to be improved to a reasonable scientific standard.
7. What are the future research directions and limitations of this study?
8. In summary, the work has the potential to be published but before it should be considered for publication, it has to pass through professional proofreading and all the highlighted points above need to be corrected and implemented.
Author Response
Review 1: The English language needs more work. There are many grammatical and typo mistakes in this manuscript. The paper needs to be edited by a native English speaker.
Response 1: We thank you very much for reminding us of this useful point! We are sorry for the problem with the incorrect expression. According to your suggestions, we have replaced the grammatical and typo mistakes.
Review 2: I suggest the authors revise the introduction of the study per the comments raised. The authors can also use the following points below as a guideline to help them come out with an interesting introduction that is more scientific.
Background & Significance: (What general background does the reader need to understand the manuscript and how important is it in the context of scientific research).
Problem definition: (What are the research questions to fill in the gaps of the existing knowledge body or methodology (Methods are not a contribution, but a tool to assess whether your hypothesis or predictions are supported or not supported)? I would like to see well-developed arguments for predicting or proposing specific relationships in this study.
Motivations & Objectives: (Why are you conducting the study and what are the goals to achieve?)
Response 2: Thank you for your careful review! We have modified the parts of the “Introduction”. We describe the background and motivation of the paper in detail. All the modifications were highlighted in the “Introduction”.
Review 3: Explain the variable selection procedure and why the authors choose these variables for the study.
Response 3: We thank you very much for reminding us of this useful point! According to your suggestions, we have modified Part 4 and introduced the procedure of variable selection in that part.
Review 4: I would like to suggest that authors should update the introduction, literature, and results part. Specifically, the latest research trends, and to highlight the academic frontier of the research, the references of the recent year need to be referenced.
Response 4: We thank you very much for reminding us of this useful point! According to your suggestions, we have updated the “Introduction” and “Experiments”. And we added references for the recent year.
Review 5: The author(s) need to compare their results (Each Finding) with past studies (what was provided in the article is not compares of results but an explanation of views from past authors) and in comparing the result from the empirical investigations the author(s) should as much as possible provide a recast of the comparison made and the supposed implications or advantages of the new finding made with those discovered by past authors. This will ensure justice to the extant literature and also manifest the superiority of the current findings over the past findings.
Response 5: We thank you very much for reminding us of this useful point! According to your suggestions, we provided the analysis of the result in part “Experiments”.
Review 6: The study's conclusion should have a realistic empirical overview and not a summary. The conclusion should provide an overall thought from the author(s) empirical and conceptual viewpoint on why and how things exist or went the way they were discovered, what are the implications of that and what are the advantages to the key study areas under survey. Is this development attributable to the method or the variables or are these findings the reality of the situations on the ground or what is to be expected soonest etc? This should be presented in a professional, logical, and philosophical way to convey some key scientific thoughts to the readers. The present conclusion provided in the manuscript needs to be improved to a reasonable scientific standard.
Response 6: We thank you very much for reminding us of this useful point! According to your suggestions, we modified the part “Conclusion”.
Review 7: What are this study's future research directions and limitations?
Response 7: We thank you very much for reminding us of this useful point! According to your suggestions, we modified the part “Conclusion” with the added future research directions.
Review 8: In summary, the work has the potential to be published but before it should be considered for publication, it has to pass through professional proofreading and all the highlighted points above need to be corrected and implemented.
Response 8: We thank you for your careful review!

Reviewer 3 Report
see attached

Author Response
Review 1: This part is outlined but not in sufficient depth to provide comprehensive understanding for a reader who does not have expertise in the embedding methods. The technical aspects of path scoring are quantified in three quantities: question matching degree, answer completing level, and path confidence. Each of these boil down to Lp-norms of embedding quantities, which is why some understanding of the embeddings themselves is necessary to understand this work. Some comments on learning and regularization are made. In the main body, these methods are outlined relative to TransE.
Response 1: We thank you very much for reminding us of this useful point! We are sorry for the unclear expression of the embedding methods(TransE). We gave the supplementary description in the new version. All the modifications were highlighted in the paper.
Review 2: Appendix A tries to give some motivation by example for the concept of interpretability, but does not provide a technical understanding of why IMR should lead to superior experimental results. Appendix 2 outlines the technical aspects of IMR starting from other embedding methods (RotatE and ComplEx).
Response 2: We thank you very much for reminding us of this useful point! We are sorry for the unclear expression in Appendix A and Appendix 2. Appendix A is only for explaining the consistency between the interpretability of the model and its semantic meaning. And we give a detailed analysis of the experimental results to prove the model has better performance. All the modifications were highlighted in the new version of the paper.
Review 3: Several comments can be found in the marked-up manuscript where some typos are also identified
Response 3: We thank you very much for reminding us of this helpful point! We are sorry for the problem with the incorrect expression. According to your suggestions, we have modified the typo mistakes.

Reviewer 4 Report
This paper presented an approach on building a better temporal knowledge graph reasoning module on the idea of multi-hop forecasting. The idea is interesting and the given appendix A helps us to have the understanding of how the actual reasoning in the authors' scenario worked in their approach. Although the paper seems very nice when we are in 2021, the area of TKG reasoning is really actively researched area and we saw some newer ideas which appeared after that. For example, in AAAI2022, we see the paper [A] which utilized BoxEmbedding in their idea. Since this paper did not mention these newly presented comparable approaches, it is unclear the proposed idea still works better than them. Please compare to more recent works presented in the field and clarify where the proposed idea produces some advantages from them. [A] Messner, J., Abboud, R., & Ceylan, I. I. (2022). Temporal Knowledge Graph Completion Using Box Embeddings. Proceedings of the AAAI Conference on Artificial Intelligence, 36(7), 7779-7787. https://doi.org/10.1609/aaai.v36i7.20746
Author Response
Review 1: This paper presented an approach on building a better temporal knowledge graph reasoning module on the idea of multi-hop forecasting. The idea is interesting and the given appendix A helps us to have the understanding of how the actual reasoning in the authors' scenario worked in their approach. Although the paper seems very nice when we are in 2021, the area of TKG reasoning is really actively researched area and we saw some newer ideas which appeared after that. For example, in AAAI2022, we see the paper [A] which utilized Box Embedding in their idea. Since this paper did not mention these newly presented comparable approaches, it is unclear the proposed idea still works better than them. Please compare to more recent works presented in the field and clarify where the proposed idea produces some advantages from them. [A] Messner,J.,Abboud,R.,& Ceylan,I.I.(2022).Temporal Knowledge Graph Completion Using Box Embeddings. Proceedings of the AAAI Conference on Artificial Intelligence, 36(7), 7779-7787. https://doi.org/10.1609/aaai.v36i7.20746
Response 1: We thank you very much for reminding us of this useful point! We give the comparison between the model based on BoxEmbedding and our method under the same datasets. And our experiments have better performance under the widely used datasets(ICEWS14、ICEWS18). Besides, we described that the filtered setting for static KGs is not suitable for the reasoning task under the exploration setting, as mentioned in xERTR (Han et al., 2021). The time-aware filtering scheme only filters out triples that are genuine at the query time. But the model based on BoxEmbedding used different metrics to evaluate the performance. We have no access to the code of that paper. According to the limited time and words, we give a comparison about that in the next paper.

Round 2
Reviewer 4 Report
The reviewer understand the difficulties about directly comparing to the related work which was mentioned in the previous review comment. Also the related work now cited in the paper clearly in a reasonable context. While the revised version still did not include direct comparison to it, the shown conclusions with their experimentations are said to be the best one that the authors can do in the current condition.